# Cell Cycle Regulation and DNA Damage Response Networks in Diffuse- and Intestinal-Type Gastric Cancer

**DOI:** 10.3390/cancers13225786

**Published:** 2021-11-18

**Authors:** Shihori Tanabe, Sabina Quader, Ryuichi Ono, Horacio Cabral, Kazuhiko Aoyagi, Akihiko Hirose, Hiroshi Yokozaki, Hiroki Sasaki

**Affiliations:** 1Division of Risk Assessment, Center for Biological Safety and Research, National Institute of Health Sciences, Kawasaki 210-9501, Japan; hirose@nihs.go.jp; 2Innovation Center of NanoMedicine (iCONM), Kawasaki Institute of Industrial Promotion, Kawasaki 210-0821, Japan; sabina-q@kawasaki-net.ne.jp; 3Division of Cellular and Molecular Toxicology, Center for Biological Safety and Research, National Institute of Health Sciences, Kawasaki 210-9501, Japan; onoryu@nihs.go.jp; 4Department of Bioengineering, Graduate School of Engineering, University of Tokyo, Tokyo 113-0033, Japan; horacio@bmw.t.u-tokyo.ac.jp; 5Department of Clinical Genomics, National Cancer Center Research Institute, Tokyo 104-0045, Japan; kaaoyagi@ncc.go.jp; 6Department of Pathology, Kobe University of Graduate School of Medicine, Kobe 650-0017, Japan; hyoko@med.kobe-u.ac.jp; 7Department of Translational Oncology, National Cancer Center Research Institute, Tokyo 104-0045, Japan; hksasaki@ncc.go.jp

**Keywords:** cancer malignancy, cell cycle regulation, epithelial-mesenchymal transition (EMT), DNA damage response, gastric cancer, molecular network, network analysis

## Abstract

**Simple Summary:**

The epithelial-mesenchymal transition (EMT) is an important hallmark in drug resistance and cancer malignancy in cancer stem cells (CSCs). In this study, gene expression in diffuse- and intestinal-type gastric cancer (GC) was investigated to reveal the precise mechanism of EMT. The pathways of cell cycle regulation and DNA damage response were found to be altered in diffuse- and intestinal-type GC. The findings of this study may provide broader insights into CSCs, with new possibilities of the involvement of the cell cycle in EMT.

**Abstract:**

Dynamic regulation in molecular networks including cell cycle regulation and DNA damage response play an important role in cancer. To reveal the feature of cancer malignancy, gene expression and network regulation were profiled in diffuse- and intestinal-type gastric cancer (GC). The results of the network analysis with Ingenuity Pathway Analysis (IPA) showed that the activation states of several canonical pathways related to cell cycle regulation were altered. The G_1_/S checkpoint regulation pathway was activated in diffuse-type GC compared to intestinal-type GC, while canonical pathways of the cell cycle control of chromosomal replication, and the cyclin and cell cycle regulation, were activated in intestinal-type GC compared to diffuse-type GC. A canonical pathway on the role of BRCA1 in the DNA damage response was activated in intestinal-type GC compared to diffuse-type GC, where gene expression of BRCA1, which is related to G_1_/S phase transition, was upregulated in intestinal-type GC compared to diffuse-type GC. Several microRNAs (miRNAs), such as mir-10, mir-17, mir-19, mir-194, mir-224, mir-25, mir-34, mir-451 and mir-605, were identified to have direct relationships in the G_1_/S cell cycle checkpoint regulation pathway. Additionally, cell cycle regulation may be altered in epithelial-mesenchymal transition (EMT) conditions. The alterations in the activation states of the pathways related to cell cycle regulation in diffuse- and intestinal-type GC highlighted the significance of cell cycle regulation in EMT.

## 1. Introduction

In cancer stem cells (CSCs) in general, epithelial-mesenchymal transition (EMT) networks play an important role in acquiring drug resistance and malignant cancer feature [1]. Alterations in the gene expression of molecular network pathways result in phenotypic changes. Diffuse-type gastric cancer (GC) has many more mesenchymal characteristics, which is an important feature of EMT, compared with intestinal-type GC. Accordingly, the gene expression profiles have been analyzed for diffuse- and intestinal-type GC to reveal the network pathways in EMT and CSCs [2]. Our previous findings identified several molecular networks and the related microRNAs (miRNAs) in intestinal- and diffuse-type GC [2,3]. A few recent studies have revealed the regulation of non-coding RNAs, including miRNAs, in drug resistance and EMT in cancer [4,5,6]. It has also been indicated that the knockdown of circular NOP10, a circular RNA that promotes tumor metastasis and EMT, decreased the numbers of cells in the S phase and increased the number of cells in the G_2_/M phase [6].

The gene expression signature revealed that the ratio of CDH2 (N-cadherin) to CDH1 (E-cadherin) distinguishes diffuse- and intestinal-type GC [7]. The immunohistochemical expression of E-cadherin clearly discriminated diffuse- and intestinal-type GC [8]. The decreased expression of E-cadherin leads to the sparse phenotype of the cells, which represents the phenotype of EMT. The EMT programs induce CSC stemness, conferring therapeutic resistance [9]. Gastric cancer subtypes can be classified into diffuse (genomically stable; GS), intestinal (chromosomally instable; CIN), microsatellite instability-high, and Epstein-Barr virus-positive subtypes [10]. Since defects in the DNA damage response lead to genome instability and cancer progression, poly(ADP-ribose) polymerase (PARP) inhibitors have the potential to treat GC by blocking the single-strand break repair, followed by tumor cell death and synthetic lethality in homologous recombination-deficient tumor cells [11]. Immune checkpoint inhibition therapy has some limited application in GC, since anti-PD1 monoclonal antibodies, such as pembrolizumab and nivolumab, have significant responses only in minor populations of GC, such as microsatellite instability-high or Epstein-Barr virus-positive subtypes [12]. Gene expression and pathway analyses of CTNNB1 (β-catenin) identified an important role of β-catenin signaling in the regulation of stem cell pluripotency and cancer signaling [13]. Although cell proliferation and cell cycle regulation are essential for identifying potential therapeutic targets in cancer, the precise mechanism of cell cycle regulation and EMT has not been fully revealed. This article focuses on the roles of cell cycle regulation pathways in diffuse- and intestinal-type GC, as cell cycle regulation may play a critical role in intestinal- and diffuse-type GC. Consequently, the mechanism of cancer drug resistance through the involvement of the cell cycle in EMT and CSCs is highlighted, which might reveal future therapeutic potential.

## 2. Materials and Methods

### 2.1. Data Analysis of Diffuse- and Intestinal-Type GC

The RNA sequencing data of diffuse- and intestinal-type GC are publicly available in The Cancer Genome Atlas (TCGA) of the cBioPortal for Cancer Genomics database [10,14,15] at the National Cancer Institute (NCI) Genomic Data Commons (GDC) data portal [10,16]. Publicly available data on stomach adenocarcinoma in the TCGA [15] were compared between diffuse-type GC, which is genomically stable (*n* = 50), and intestinal-GC, which has a feature of chromosomal instability (*n* = 223), in TCGA Research Network publications, as previously described [2,10].

### 2.2. Network Analysis

Data on intestinal- and diffuse-type GC from the TCGA cBioPortal for Cancer Genomics were uploaded and analyzed through the use of Ingenuity Pathway Analysis (IPA) (QIAGEN Inc., Hilden, Germany) [17].

### 2.3. Data Visualization

The results of network analyses of the gene expression data on diffuse- and intestinal-type GC were visualized with Tableau software (https://www.tableau.com (accessed on 18 November 2021)).

### 2.4. Statistical Analysis

The RNA sequencing data on diffuse- and intestinal-type GC were analyzed in Student’s t-test. The z-scores of intestinal- and diffuse-type GC samples were compared, and the difference was considered to be significant at *p* < 0.00001, as previously described [2]. Activation z-score in each pathway was calculated in IPA to show the level of the activation.

## 3. Results

### 3.1. Canonical Pathways in Diffuse- and Intestinal-Type GC

Canonical pathways in diffuse- and intestinal-type GC are shown with activation z-score in Figure 1 and Table 1. Genes significantly altered in diffuse- and intestinal-type GC (2815 IDs) were analyzed by a network analysis, which identified 47 canonical pathways with absolute z-scores > 1 in the diffuse- and intestinal-type GC (Table 1). These canonical pathways included Cell cycle control of chromosomal replication, Cyclins and cell cycle regulation, and Role of BRCA1 in DNA damage response, which were activated in intestinal-type GC compared to diffuse-type GC, and Cell cycle: G_1_/S cell cycle checkpoint regulation, and Cell cycle: G_2_/M DNA damage checkpoint regulation, which were activated in diffuse-type GC compared to intestinal-type GC.

### 3.2. Cell Cycle-Related Canonical Pathways in Diffuse- and Intestinal-Type GC

#### 3.2.1. Cell Cycle Control of Chromosomal Replication was Activated in Intestinal-Type GC

The molecule activity predictor in IPA-predicted activation of the cell cycle control of chromosomal replication pathway in intestinal-type GC (Figure 2). During cell cycle progression, DNA topoisomerase II activity is controlled [18]. Analysis of the direct relationships of RNA–RNA interactions of microRNA targeting the cell cycle control of chromosomal replication pathway revealed the relationships between miRNAs and the targeted molecules (Table 2).

#### 3.2.2. The G_1_/S Cell Cycle Checkpoint Regulation Pathway was Activated in Diffuse-Type GC

The molecule activity predictor in IPA predicted the activation of the G_1_/S cell cycle checkpoint regulation pathway in diffuse-type GC (Figure 3). In the G_1_/S cell cycle checkpoint regulation pathway, DNA damage induces p53, which is expected to be activated in diffuse-type GC. Analysis of the direct relationships of RNA–RNA interactions in the G_1_/S cell cycle checkpoint regulation pathway revealed non-coding RNAs, including nine miRNAs and a biologic drug (MYC-targeting siRNA DCR-MYC) (Table 3).

#### 3.2.3. The Cyclin and Cell Cycle Regulation Pathway was Activated in Intestinal-Type GC

Molecule activity predictor in IPA predicted activation of the cyclins and cell cycle regulation pathway in intestinal-type GC (Figure 4). A study has reported that prolyl 4-hydroxylase subunit alpha 1, which regulates cyclin-dependent kinases (CDKs), cyclins, and CDK inhibitors, was associated with EMT, tumor cell invasion and metastasis of lung adenocarcinoma [19]. An analysis of the direct relationships of RNA–RNA interactions of microRNA targeting in the cyclins and cell cycle regulation pathway revealed the relationships between miRNAs and the targeted molecules (Table 4).

#### 3.2.4. Canonical Pathway of the Role of BRCA1 in the DNA Damage Response was Activated in Intestinal-Type GC Compared to Diffuse-Type GC

The prediction of molecule activity predictor in IPA demonstrated that the canonical pathway on the role of BRCA1 in the DNA damage response was activated in intestinal-type GC compared to diffuse-type GC (Figure 5). The role of BRCA1 in the DNA damage response pathway was identified as the most significant canonical pathway, with a *p*-value of 6.6 × 10^−12^. Gene expression of BRCA1, which is associated with G_1_/S transition, increased in intestinal-type GC. BRCA1 encodes a 190 kD nuclear phosphorylation protein, which maintains genomic stability and functions as a tumor suppressor. It is interesting that p53 and c21CIP1 are activated in intestinal-type GC in the role of BRCA1 in the DNA damage response pathway. BRCA1 may be involved in the activation of p53. An analysis of the direct relationships of RNA–RNA interactions of microRNA targeting the role of BRCA1 in the DNA damage response pathway revealed the relationships between miRNAs and the targeted molecules (Figure 5, Table 5).

#### 3.2.5. The G_2_/M DNA Damage Cell Cycle Checkpoint Regulation Pathway in Diffuse- and Intestinal-Type GC

The G_2_/M DNA damage cell cycle checkpoint regulation pathway was identified as a related canonical pathway in diffuse- and intestinal-type GC (Figure 6). Analysis of the direct relationships of RNA–RNA interactions in the G_2_/M DNA damage cell cycle checkpoint regulation pathway revealed non-coding RNAs, including nine miRNAs and a biological drug (lipid-encapsulated anti-PLK1 siRNA TKM-080301) (Table 6).

## 4. Discussion

The network analysis of gene expression data on diffuse- and intestinal-type GC revealed that the activation state of canonical pathways related to cell cycle regulation, including cell cycle control of chromosomal replication, G_1_/S cell cycle checkpoint regulation, cyclins and cell cycle regulation, and G_2_/M DNA damage cell cycle checkpoint regulation pathways, altered in diffuse- and intestinal-type GC. A previous study showed that G_1_/S arrest induces EMT by ribosome biogenesis [20]. This finding of G_1_/S arrest in EMT may be associated with the finding in the current study in terms of activating the G_1_/S cell cycle checkpoint regulation pathway in diffuse-type GC. SMAD4, which was activated in diffuse-type GC, is involved in EMT [21]. The silencing of SMAD4 reversed EMT in hepatocytes [22]. The analyses on RNA-RNA relationships highlighted that miRNAs had direct interactions in pathways related to cell cycle regulation. mir-10 (microRNA 99a), mir-17 (microRNA 17), mir-19 (microRNA 19a), mir-194 (microRNA 194-1), mir-224 (microRNA 224), mir-25 (microRNA 25), mir-34 (microRNA 34a), mir-451 (microRNA 451a), mir-605 (microRNA 605), and MYC-targeting siRNA DCR-MYC were identified as molecules having direct relationships (RNA–RNA interactions) in G_1_/S cell cycle checkpoint regulation pathway. It has been previously revealed that the miR-17/20 cluster is transcriptionally regulated by MYC, E2F, and cyclin D1, and miR-34 is a direct transcriptional target of p53 and regulates the cell cycle [23]. The cell cycle regulation of miRNAs can be involved in cancer. Small nucleolar RNA host gene 7 (SNHG7), an oncogenic long non-coding RNA, promotes cell migration via the miR-34a-Snail-EMT axis in gastric cancer [24]. The involvement of miR-34 in EMT and GC is consistent with the results showing the direct relationship between miR-34 and the G_1_/S cell cycle checkpoint regulation pathway, which is activated in diffuse-type GC. Another study suggested that miRNA-33a inhibited EMT and the metastasis of GC via the Snail/Slug pathway, which may be related to the miRNA regulation of EMT in intestinal-type GC [25]. MYC-targeting siRNA, a biologic drug targeting MYC, has been identified to have direct relationships in the G_1_/S cell cycle checkpoint regulation pathway. It was shown previously that the silencing of Aurora kinase B, which is upregulated in GC, decreased the expression of MYC, arrested the cell cycle in the G_2_/M phase and inhibited the migration of GC [26]. MYC-targeting siRNA might also be involved in regulation of the cell cycle and EMT.

The current study highlights the cell cycle regulation and non-coding RNA regulation in diffuse- and intestinal-type GC. A long non-coding RNA, LINC00460, induced EMT, and cell proliferation and metastasis in head and neck squamous cell carcinoma via translocation of peroxiredoxin-1 into the cell nucleus [27]. Non-coding RNAs, such as microRNAs, have various roles in biological and pathological processes, including the cell cycle, proliferation, EMT, and drug resistance [28]. This study has the potential to identify the target non-coding RNAs for cancer therapy, where the non-coding RNAs have roles as anti-cancer drugs or the targets of anti-cancer drugs. The knowledge gaps include differences in the subtypes of tumor cells in terms of regulation of the cell cycle and EMT, and the influence of the cancer microenvironment on RNA interactions. Researchers have been trying to unravel the RNA networks in cancer subtypes and microenvironments through comprehensive analyses including microarrays, RNA sequencing, genome-wide sequencing, and proteomic analyses of the tumor tissue or cell population, or even a single cell [29,30,31,32,33]. The problem still remains regarding the communication between cells in the orchestrated organs and tissues in human body. In the next five years, rapid advances in artificial intelligence, 3D organoids, and organ modeling may realize a harmonized android of the human body. Investigation into RNA regulation and EMT from the point of view of anti-cancer drug resistance and identification of therapeutic targets would be the future research direction.

The current study has some limitations, as the data of diffuse- and intestinal-type GC have been compared only in terms of cell cycle regulation. To reveal the detailed mechanisms of cell cycle regulation of different subtypes of GC and the potential applicability of targeted therapy, an analysis of the microsatellite instability-high and Epstein-Barr virus-positive subtypes would be needed.

## 5. Conclusions

Activation stated of canonical pathways related to cell cycle regulation altered in diffuse- and intestinal-type GC. The canonical pathways on G_1_/S cell cycle checkpoint regulation were activated in diffuse-type GC, while canonical pathways on cell cycle control of chromosomal replication and cyclins and cell cycle regulation were activated in intestinal-type GC. The canonical pathway on role of BRCA1 in DNA damage response was activated in intestinal-type GC compared with diffuse-type GC. Some molecules, such as SMAD4 in the G_1_/S cell cycle checkpoint regulation pathway, are involved in the regulation of EMT. Cell cycle regulation may also be altered under EMT conditions in diffuse-type GC. The findings of the study showing how the activation states of the canonical pathways related to cell cycle regulation altered in diffuse- and intestinal-type GC highlighted the involvement of cell cycle regulation in cancer malignancy. The precise mechanisms of how the compartment molecules in the cell cycle pathways regulate EMT and CSCs would be beneficial for future investigation.

## Figures and Tables

**Figure 1 cancers-13-05786-f001:**
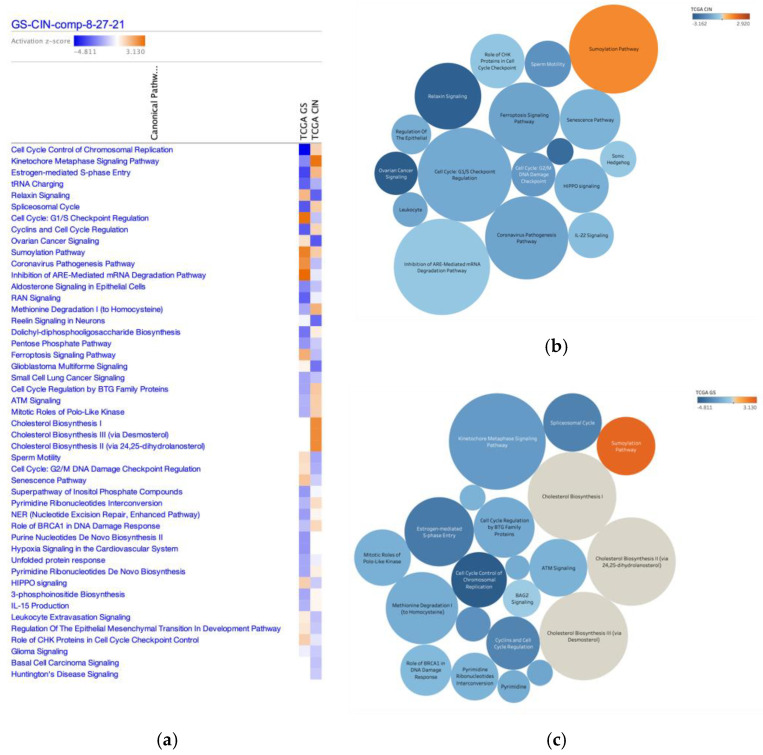
Canonical pathways altered in diffuse- and intestinal-type GC. (**a**) Canonical pathways with an absolute z-score of >1 for diffuse- and intestinal-type GC are shown (TCGA GS; diffuse-type GC, TCGA CIN; intestinal-type GC) (**b**). The size of each circle shows the activation score in diffuse-type GC. Color indicates the activation score in intestinal-type GC. (**c**) The size of each circle indicates the activation z-score for intestinal-type GC. Color indicates the activation z-score of intestinal-type GC.

**Figure 2 cancers-13-05786-f002:**
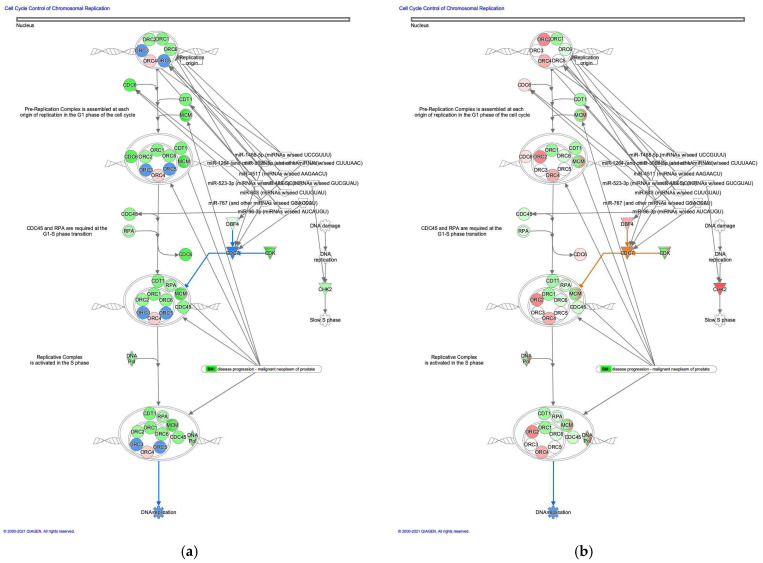
Cell cycle control of chromosomal replication was activated in intestinal-type GC. (**a**) Gene expression and pathway activity predictions in diffuse-type GC are shown. (**b**) Gene expression and pathway activity predictions in intestinal-type GC are shown. The genes whose expression was altered in diffuse- and intestinal-type GC are shown in pink (upregulated) or green (downregulated). Predicted activation and inhibition are shown in orange and blue, respectively.

**Figure 3 cancers-13-05786-f003:**
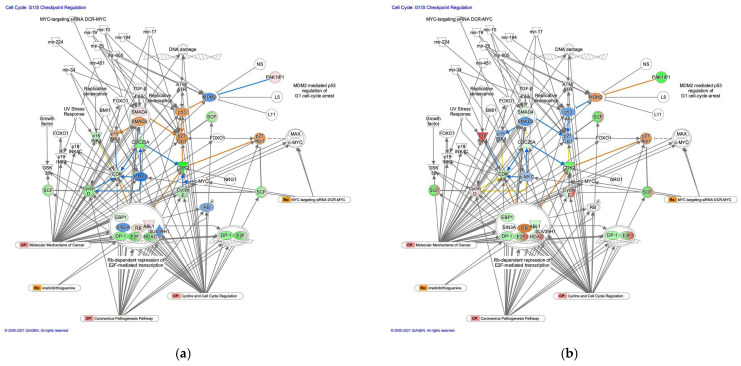
The G_1_/S cell cycle checkpoint regulation pathway was activated in diffuse-type GC. (**a**) Gene expression and pathway activity predictions in diffuse-type GC are shown. (**b**) Gene expression and pathway activity predictions in intestinal-type GC are shown. The genes whose expression was altered in diffuse- and intestinal-type GC are shown in pink (upregulated) or green (downregulated). Predicted activation and inhibition are shown in orange and blue, respectively.

**Figure 4 cancers-13-05786-f004:**
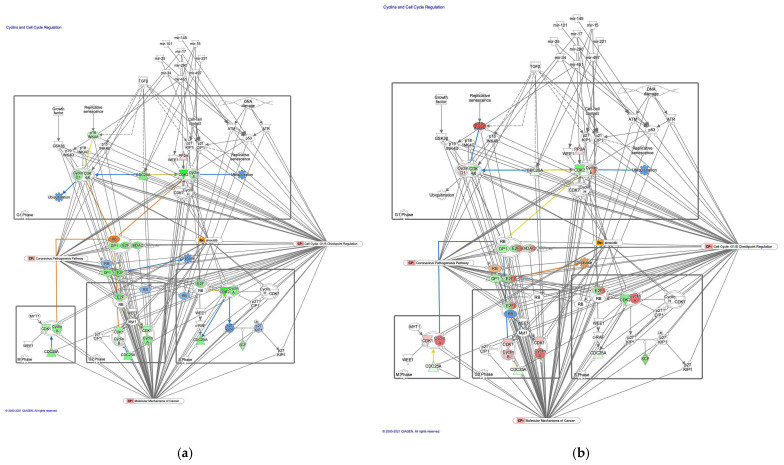
The cyclin and cell cycle regulation pathway was activated in intestinal-type GC. (**a**) Gene expression and pathway activity predictions in diffuse-type GC are shown. (**b**) Gene expression and pathway activity predictions in intestinal-type GC are shown. The genes whose expression was altered in diffuse- and intestinal-type GC are shown in pink (upregulated) or green (downregulated). Predicted activation and inhibition are shown in orange and blue, respectively.

**Figure 5 cancers-13-05786-f005:**
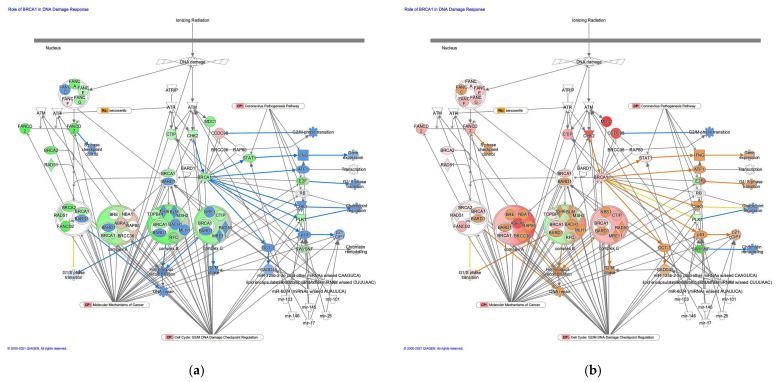
The role of BRCA1 in the DNA damage response was activated in intestinal-type GC. (**a**) Gene expression and pathway activity predictions in diffuse-type GC are shown. (**b**) Gene expression and pathway activity predictions in intestinal-type GC are shown. The genes whose expression was altered in diffuse- and intestinal-type GC are shown in pink (upregulated) or green (downregulated). Predicted activation and inhibition are shown in orange and blue, respectively.

**Figure 6 cancers-13-05786-f006:**
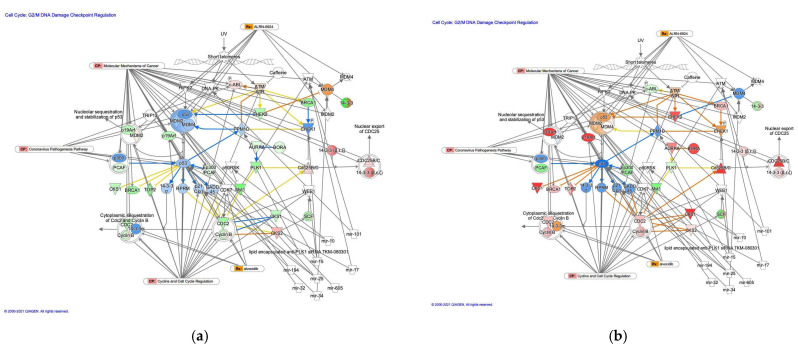
The G_2_/M DNA damage cell cycle checkpoint regulation pathway in diffuse- and intestinal-type GC. (**a**) Gene expression and pathway activity predictions in diffuse-type GC are shown. (**b**) Gene expression and pathway activity predictions in intestinal-type GC are shown. The genes whose expression was altered in diffuse- and intestinal-type GC are shown in pink (upregulated) or green (downregulated). Predicted activation and inhibition are shown in orange and blue, respectively.

**Table 1 cancers-13-05786-t001:** Canonical pathways altered in diffuse- and intestinal-type GC. The pathways are sorted in the order of the activation z-scores.

Canonical Pathways	Diffuse-Type GC	Intestinal-Type GC
Cell Cycle Control of Chromosomal Replication	−4.811	0.962
Kinetochore Metaphase Signaling Pathway	−2.271	2.92
Estrogen-mediated S-phase Entry	−3.5	1.5
Relaxin Signaling	1.5	−3
tRNA Charging	−3	−1.5
Spliceosomal Cycle	−3.207	1.069
Cell Cycle: G_1_/S Checkpoint Regulation	2.982	−1.147
Cyclins and Cell Cycle Regulation	−3.13	0.894
Ovarian Cancer Signaling	0.632	−3.162
Sumoylation Pathway	2.673	1.069
Coronavirus Pathogenesis Pathway	2.335	−1.257
Inhibition of ARE-Mediated mRNA Degradation Pathway	3.13	−0.447
Aldosterone Signaling in Epithelial Cells	−2.309	−1.155
RAN Signaling	−3	−0.333
Methionine Degradation I (to Homocysteine)	−1.633	1.633
Reelin Signaling in Neurons	−0.218	−2.837
Dolichyl-diphosphooligosaccharide Biosynthesis	−2.646	0.378
Pentose Phosphate Pathway	−2	−1
Ferroptosis Signaling Pathway	1.706	−1.279
Glioblastoma Multiforme Signaling	0.243	−2.668
Cell Cycle Regulation by BTG Family Proteins	−1.732	1.155
Small-Cell Lung Cancer Signaling	−1.732	−1.155
ATM Signaling	−1.46	1.043
Mitotic Roles of Polo-Like Kinase	−1.5	1
Cholesterol Biosynthesis III (via Desmosterol)	0	2.449
Cholesterol Biosynthesis I	0	2.449
Cholesterol Biosynthesis II (via 24,25-dihydrolanosterol)	0	2.449
Sperm Motility	0.728	−1.698
Cell Cycle: G_2_/M DNA Damage Checkpoint Regulation	0.655	−1.528
Senescence Pathway	1.234	−0.926
Superpathway of Inositol Phosphate Compounds	−1.976	−0.18
Pyrimidine Ribonucleotides Interconversion	−1.414	0.707
NER (Nucleotide Excision Repair, Enhanced Pathway)	−1.877	0.209
Role of BRCA1 in DNA Damage Response	−1.225	0.816
Unfolded Protein Response	−1.667	−0.333
Purine Nucleotides De Novo Biosynthesis II	−2	0
HIPPO signaling	1	−1
Pyrimidine Ribonucleotides De Novo Biosynthesis	−1.667	0.333
Hypoxia Signaling in the Cardiovascular System	−2	0
3-Phosphoinositide Biosynthesis	−1.671	0.186
IL-15 Production	−1.46	0.209
Leukocyte Extravasation Signaling	0.408	−1.225
Regulation of the Epithelial–Mesenchymal Transition in Development Pathway	0.535	−1.069
Role of CHK Proteins in Cell Cycle Checkpoint Control	1	−0.5
Glioma Signaling	−0.333	−1
Basal Cell Carcinoma Signaling	0	−1.155
Huntington’s Disease Signaling	0	−1

**Table 2 cancers-13-05786-t002:** Direct relationships of miRNAs and targeted molecules in cell cycle control of chromosomal replication.

miRNA	Targeted Molecules	Entrez Gene Name
miR-1264 (and other miRNAs with the seed AAGUCUU)	CDC7	cell division cycle 7
	ORC6	origin recognition complex subunit 6
miR-1468-5p (miRNAs with the seed UCCGUUU)	ORC4	origin recognition complex subunit 4
	ORC6	origin recognition complex subunit 6
mir-192	CDC7	cell division cycle 7
miR-302b-5p (and other miRNAs with the seed CUUUAAC)	DBF4	DBF4 zinc finger
miR-4511 (miRNAs with the seed AAGAACU)	ORC5	origin recognition complex subunit 5
miR-489-5p (miRNAs with the seed GUCGUAU)	CHK2	checkpoint kinase 2
miR-523-3p (miRNAs with the seed AACGCGC)	CDC6	cell division cycle 6
miR-643 (miRNAs with the seed CUUGUAU)	ORC2	origin recognition complex subunit 2
	ORC5	origin recognition complex subunit 5
miR-767 (and other miRNAs with the seed GCACCAU)	CDC6	cell division cycle 6
	CDC7	cell division cycle 7
	ORC6	origin recognition complex subunit 6
miR-96-3p (miRNAs with the seed AUCAUGU)	CDC45	cell division cycle 45
	CDT1	chromatin licensing and DNA replication factor 1
	ORC4	origin recognition complex subunit 4

**Table 3 cancers-13-05786-t003:** Non-coding RNAs which have direct relationships in the G_1_/S cell cycle checkpoint regulation pathway.

Symbol	Entrez Gene Name	Location	Family	Interacting Molecules
mir-10	microRNA 99a	Cytoplasm	microRNA	SMAD4, SUV39H1, p53
mir-17	microRNA 17			Cyclin D, RB, p21Cip1
mir-19	microRNA 19a			SMAD4, p21Cip1
mir-194	microRNA 194-1			MDM2
mir-224	microRNA 224			SMAD4
mir-25	microRNA 25			MDM2, p21Cip1, p53
mir-34	microRNA 34a			CDK4/6, c-MYC, p53
mir-451	microRNA 451a			p19INK4
mir-605	microRNA 605			MDM2
MYC-targeting siRNA DCR-MYC		Other	Biologic drug	c-Myc

**Table 4 cancers-13-05786-t004:** Direct relationships of miRNAs and targeted molecules in cyclin and cell cycle regulation.

miRNA	Targeted Molecules	Entrez Gene Name
mir-10	ATM	ATM serine/threonine kinase
mir-145	p53	tumor protein p53
mir-15	CDC25A	cell division cycle 25A
	WEE1	WEE1 G2 checkpoint kinase
	c-RAF	Raf-1 proto-oncogene, serine/threonine kinase
mir-17	ATM	ATM serine/threonine kinase
	CyclinD1	
	RB	RB transcriptional corepressor 1
	p21CIP1	cyclin dependent kinase inhibitor 1A
mir-221	p27KIP1	cyclin dependent kinase inhibitor 1B
mir-25	p21CIP1	cyclin dependent kinase inhibitor 1A
	p53	tumor protein p53
mir-290	CDK2	cyclin dependent kinase 2
mir-34	CDK4/6	cyclin dependent kinase 4/6
	p53	tumor protein p53
mir-451	p19INK4D	cyclin dependent kinase inhibitor 2D
mir-497	CDC25A	cell division cycle 25A
	c-RAF	Raf-1 proto-oncogene, serine/threonine kinase

**Table 5 cancers-13-05786-t005:** Direct relationships of miRNAs and targeted molecules in the role of BRCA1 in the DNA damage response.

miRNA	Targeted Molecules	Entrez Gene Name
miR-125b-2-3p (and other miRNAs with the seed CAAGUCA)	p53	tumor protein p53
miR-302b-5p (and other miRNAs with the seed CUUUAAC)	BARD1	BRCA1 associated RING domain 1
	CTIP	RB binding protein 8, endonuclease
	GADD45	growth arrest and DNA damage inducible alpha
miR-6074 (miRNAs with the seed AUAUUCA)	FANCF	FA complementation group F
	IFNG	interferon gamma
	NBS1	nibrin
mir-101	ATM	ATM serine/threonine kinase
mir-103	p53	tumor protein p53
mir-145	p53	tumor protein p53
mir-146	STAT1	signal transducer and activator of transcription 1
mir-17	ATM	ATM serine/threonine kinase
	RB	RB transcriptional corepressor 1
	p21CIP1	cyclin dependent kinase inhibitor 1A
mir-25	p21CIP1	cyclin dependent kinase inhibitor 1A
	p53	tumor protein p53

**Table 6 cancers-13-05786-t006:** Non-coding RNAs which have direct relationships in the G_2_/M DNA damage cell cycle checkpoint regulation pathway.

Symbol	Entrez Gene Name	Location	Family	Interacting Molecules
Lipid-encapsulated anti-PLK1 siRNA TKM-080301		Other	Biologic drug	PLK1
mir-10	microRNA 99a	Cytoplasm	microRNA	p53, p90RSK
mir-101	microRNA 101-1			ATM, DNA-PK
mir-15	microRNA 15a			CHEK1, PPM1D, WEE1
mir-17	microRNA 17			ATM, p21Cip1
mir-194	microRNA 194-1			MDM2
mir-25	microRNA 25			MDM2, p21Cip1, p53
mir-32	microRNA 32			MDM2
mir-34	microRNA 34a			MDM4, p53
mir-605	microRNA 605			MDM2

## Data Availability

In this research, the RNA sequencing data of diffuse- and intestinal-type GC, which are publicly available from The Cancer Genome Atlas (TCGA) of the cBioPortal for Cancer Genomics database [10,14,15] and from the National Cancer Institute (NCI) Genomic Data Commons (GDC) data portal, were analyzed [16]. The publicly available data of stomach adenocarcinoma from TCGA (NCI, USA: https://www.cancer.gov/about-nci/organization/ccg/research/structural-genomics/tcga (accessed on 18 November 2021)) [15], and intestinal- and diffuse-type GC data, which are noted as having chromosomal instability (CIN) (*n* = 223) and being genomically stable (GS) (*n* = 50), respectively, from TCGA Research Network publications, were compared [10].

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
