# Peer review of "Cell Cycle Regulation and DNA Damage Response Networks in Diffuse- and Intestinal-Type Gastric Cancer"

_cancers, 2021, doi:10.3390/cancers13225786_

Round 1

Reviewer 1 Report

Dear Authors,

The manuscript looks much better now, However it can still be improved.

Reviewer 2 Report

The authors modified the paper according to our suggestions.

We recommend acceptance.

This manuscript is a resubmission of an earlier submission. The following is a list of the peer review reports and author responses from that submission.

Round 1

Reviewer 1 Report

Shihori Tanabe et.al, observed role of Cell Cycle Regulation and DNA Damage Response Networks in Diffuse-and Intestinal-Type Gastric Cancer by using existing IPA pathway analysis tool. Article is poorly written, inadequate materials and methods and  interpretation of data is also inadequate.

  1. Introduction poorly written, authors should elaborate the issue how cell EMT and CSC plays role to acquire drug resistance.
  2. Cancer is heterogeneous, and different cells has in different states of cell cycle, and it is hard to say cell cycle is influencing the drug resistance (line from 64 to 71). Please provide rationale to explain further, it is interesting statement although.
  3. Material and Methods, again it is poorly written. By using one tool to say cell cycle stage is influencing the EMT transition eventually acquiring cancer resistance is hard to interpret the data.
  4. To further support authors claim on cell cycle stage influence on EMT transition and CSC eventually drug resistance. Please compare different cancer models.
  5. Results 3.1 canonical pathways in Diffuse- and Intestine-type GC: authors wrote just what is there in the figures, there is no explanation for anything and nothing is there to understand further. Better to rewrite with the explanation by comparing the existing data with your novel observation or claim.    
  6. 2.1 Cell cycle control of chromosomal replication was activated in intestinal-type GC:

Hard to understand this statement from the figure 2, authors should explain in detail by comparing diffuse and intestinal GC, how cell cycle control of chromosomal replication. Authors can refer existing literature and quote them.

  1. Table 2. Direct relationships of miRNAs and targeted molecules in Cell cycle control of chromosomal replication:

Different microRNAs targeting some molecules, apart from this no one can understand if authors use some molecules. It is just data copy paste from the analysis no formatting also. At least authors can use scientific terms genes, mRNAs like that. And try to explain From molecules to To Molecules what it is. At least elaborate them (CDC7=?, ORC6=? And so on..) what they stand for.

  1. 2.2. Cell cycle: G1/S checkpoint regulation pathway was activated in diffuse-type GC

: Again, hard to understand, it is figure explanation not the data interpretation

  1. Table 3. Non-coding RNAs which have direct relationships in Cell cycle: G1/S checkpoint regulation pathway. Why this table, there is some microRNAs symbols and their entrez gene name, apart from this nothing can be understood. Please explain where these micro RNAs were targeting, just like Table2 to understand.

It is good that authors have novel finding, please explain them by referring the existing data.

  1. 2.3. Cyclins and cell cycle regulation pathway was activated in intestinal-type GC: Figure legend explanation no broader explanation here also.
  2. Table 4. Direct relationships of miRNAs and targeted molecules in Cyclins and cell cycle regulation

Set of microRNAs targeted some genes, Okay, whether this already reported or these microRNAs were entirely novel. If these microRNA and their target genes reported please use their references.

  1. 2.4. Role of BRCA1 in DNA damage response pathway was activated in intestinal-type GC:

It is interesting observation, that BRCA1 is pathway is active even in basal cells itself. Better explanation is required. Hard to interpret data from the figures.

  1. 2.5. Cell cycle: G2/M DNA damage checkpoint regulation pathway in diffuse- and intestinal- type GC

Again, same issue it is figure legend, no explanation for what authors claim is.

  1. Table 6. Direct relationships of miRNAs and targeted molecules in Cell cycle: G2/M DNA damage checkpoint regulation pathway

These are novel microRNAs targeting these genes or they were existing in literature, if they were reported please refer those articles for further reference.

Reviewer 2 Report

The present study by Shihori Tanabe et al entitled “Cell Cycle Regulation and DNA Damage Response Networks 2 in Diffuse-and Intestinal-Type Gastric Cancer“ discusses about the EMT networks and their role in acquiring drug resistance and malignant features in cancer stem cells. They have analyzed the expression profiles of diffuse type and intestinal type gastric cancer and also revealed the network pathways in EMT and CSCs. • This study is just prediction based and analysis has been done on publicly available database. • First of what is the rationale of this study. • Whether this the first study of this type or this is just an additional study. • There are various grammatical errors. • There is no link from one section to another. • Authors should at least do some staining in some tissue samples to show that there is pathological differences between the two types of cancer. • Authors also mention about ration of N-Cadherin and E-Cadherin, please provide the immunohistochemical staining of those proteins in some patient samples.

Reviewer 3 Report

Dear Editor, thank you so much for inviting me to revise this manuscript about gastric cancer.

This study addresses a current topic.

The manuscript is quite well written and organized. English could be improved.

Figures and tables are comprehensive and clear.

The introduction explains in a clear and coherent manner the background of this study.

We suggest the following modifications:

  • Introduction section: although the authors correctly included important papers in this setting, we believe a couple of studies should be cited within the introduction ( PMID: 34167572; PMID: 33508962), only for a matter of consistency. We think it might be useful to introduce the topic of this interesting study.
  • Methods and Statistical Analysis: nothing to add.
  • Discussion section: Very interesting and timely discussion. Of note, the authors should expand the Discussion section, including a more personal perspective to reflect on. For example, they could answer the following questions – in order to facilitate the understanding of this complex topic to readers: what potential does this study hold? What are the knowledge gaps and how do researchers tackle them? How do you see this area unfolding in the next 5 years? We think it would be extremely interesting for the readers.

However, we think the authors should be acknowledged for their work. In fact, they correctly addressed an important topic in this setting, the methods sound good and their discussion is well balanced.

One additional little flaw: the authors could better explain the limitations of their work, in the last part of the Discussion.

We believe this article is suitable for publication in the journal although some revisions are needed. The main strengths of this paper are that it addresses an interesting and very timely question and provides a clear answer, with some limitations.

We suggest a linguistic revision and the addition of some references for a matter of consistency. Moreover, the authors should better clarify some points.